# Recent Progress of Exosomes in Multiple Myeloma: Pathogenesis, Diagnosis, Prognosis and Therapeutic Strategies

**DOI:** 10.3390/cancers13071635

**Published:** 2021-04-01

**Authors:** Xi Wang, Lin He, Xiaobing Huang, Shasha Zhang, Wanjun Cao, Feifei Che, Yizhun Zhu, Jingying Dai

**Affiliations:** 1Sichuan Academy of Medical Sciences & Sichuan Provincial People’s Hospital, School of Medicine of University of Electronic Science and Technology of China, No. 32, West Section 2, First Ring Road, Qingyang District, Chengdu 610000, China; xiwang20210322@126.com (X.W.); helin514@126.com (L.H.); huangxiaobing@med.uestc.edu.cn (X.H.); shashazhang2021@126.com (S.Z.); feifeiche2021@126.com (F.C.); 2School of Pharmacy, North Sichuan Medical College, Nanchong 637000, China; wanjun219cwj@126.com; 3State Key Laboratory of Quality Research in Chinese Medicine & School of Pharmacy, Macau University of Science and Technology, Avenida Wai Long, Taipa, Macau 999078, China

**Keywords:** exosomes, multiple myeloma, biological nanocarrier, therapeutic strategies

## Abstract

**Simple Summary:**

In the pathogenesis of multiple myeloma (MM), some exosomes act on different cells in the bone marrow microenvironment, which can create an environment conducive to the survival and growth of MM cells. In addition, due to the abnormal expression of cargos in the exosomes of MM patients, exosomes may help with the diagnosis and prognosis of MM. In contrast to traditional nanomaterials, exosomes exhibit very good safety, biocompatibility, stability and biodegradability, which shows their potential for delivering anti-cancer drugs and cancer vaccines. Given the research in recent decades, exosomes are becoming increasingly relevant to MM. Although exosomes have not been applied in the clinic for help with diagnosing, prognosticating or providing therapy for MM, they are very promising for clinical applications concerning MM, which will possibly materialize in the near future. Therefore, this review is worth reading for further understanding of the important roles of exosomes in MM..

**Abstract:**

Multiple myeloma (MM) is a hematological malignancy that is still incurable. The bone marrow microenvironment (BMM), with cellular and non-cellular components, can create a favorable environment for the survival, proliferation and migration of MM cells, which is the main reason for the failure of MM therapies. Many studies have demonstrated that exosomes play an important role in the tumor-supportive BMM. Exosomes are nanoscale vesicles that can be released by various cells. Some exosomes contribute to the pathogenesis and progression of MM. MM-derived exosomes act on different cells in the BMM, thereby creating an environment conducive to the survival and growth of MM cells. Owing to the important roles of exosomes in the BMM, targeting the secretion of exosomes may become an effective therapeutic strategy for MM. In addition, the abnormal expression of “cargos” in the exosomes of MM patients may be used to diagnose MM or used as part of a screen for the early prognoses of MM patients. Exosomes also have good biological properties, including safety, biocompatibility, stability and biodegradability. Therefore, the encapsulation of anti-cancer drugs in exosomes, along with surface modifications of exosomes with targeting molecules, are very promising strategies for cancer therapies—particularly for MM. In addition, DC-derived exosomes (DC-EXs) can express MHC-I, MHC-II and T cell costimulatory molecules. Therefore, DC-EXs may be used as a nanocarrier to deliver cancer vaccines in MM. This review summarizes the recent progress of exosome research regarding the pathogenesis of, diagnosis of, prognosis of and therapeutic strategies for MM.

## 1. Introduction

Multiple myeloma (MM) is a hematological malignancy characterized by the accumulation of clonal malignant plasma cells in the bone marrow (BM). MM accounts for about 10% of hematological malignancies. The incidence in men is higher than that in women. The median age of MM patients is about 65 years, and fewer than 3% of patients are under 40 years old [1]. The main clinical manifestations of MM are anemia, bone pain and pathological fractures, repeated infections, hypercalcemia and renal failure. Despite the significantly improved therapeutic strategies for MM, such as proteasome inhibitors, immunomodulatory drugs (IMiDs), autologous stem cell transplantation (ASCT) and cellular immunotherapy, the disease remains incurable, owing to drug resistance and frequent relapses. Therefore, investigations of the pathogenesis of, diagnosis of, prognosis of and therapeutic strategies for MM are still only on the way to making it a curable disease. In recent years, the roles of exosomes have emerged in relation to the pathogenesis, diagnosis, prognosis and therapy of MM; they show promising potential for clinical strategies against this malignancy.

## 2. Overview of Exosomes

It is known that cells are able to secrete various types of extracellular vesicles (EVs). According to the sizes, contents and formation mechanisms of EVs, they can be divided into different subgroups [2,3], including exosomes, microvesicles and apoptotic bodies. The diameters of apoptotic bodies (800–5000 nm) and microvesicles (200–1000 nm) are larger than those of exosomes (30–150 nm) [4]. Therefore, exosomes are small vesicles with a membrane structure at the nano level [5]. They carry physiological and pathological information, and efficiently transfer information between cells. With their advantages in size, biocompatibility and safety, exosomes, in some of the current research projects, perform better than liposomes in hydrophilic and hydrophobic drug delivery, although liposomes are currently the most promising nanocarriers for drug delivery with successful applications in the clinic.

### 2.1. Biogenesis of Exosomes

Exosomes are EVs of endocytic origin that can be released by various types of cells. To be specific, the biogenesis of exosomes can be divided into three stages (Figure 1): (1) Endocytic vesicles are formed by the plasma membrane. The fusion of endocytic vesicles leads to the formation of early endosomes, which then mature into late endosomes. (2) The inward budding of the late endosome membranes leads to the formation of intraluminal vesicles (ILVs). The accumulations of ILVs in the late endosomes are termed multivesicular bodies (MVBs). (3) The fusion of MVBs with the plasma membrane results in the release of ILVs, known as exosomes. [6,7]. MVBs may fuse with lysosomes for degradation, or they may fuse with the plasma membrane, releasing ILVs to the extracellular space [8]. Rab27A and Rab27B, members of the Rab family, play an important role in the release of exosomes. They can induce the transfer of the MVBs to the cellular periphery, which finally results in the fusion of MVBs with the plasma membrane [9]. At present, there are two known pathways for the formation of MVBs. The most detailed description of the process is the endosomal sorting complexes required for transport (ESCRT) pathway, which is induced by ESCRT complexes composed of four soluble multi-protein complexes (ESCRT-0, ESCRT-I, ESCRT-II and ESCRT-III) [10]. The ESCRT pathway can be figuratively seen as a “cargo” identification and membrane deformation machine. ESCRT complexes firstly recognize and classify the ubiquitinated cargo. Then, ESCRT complexes combine with the ubiquitinated cargo at the endosomes, causing the budding of the endosomal-restricted membrane to form MVBs. In addition, they can also mediate the transport of MVBs to the plasma membrane or lysosome [11]. Another pathway is ESCRT independent, the molecules involved in which include lipids, tetraphosphates and heat shock proteins [12].

### 2.2. The Composition of Exosomes

The structure of the exosome is similar to that of the liposome, consisting of a lipid bilayer membrane. Exosomes, with a diameter of 30–150 nm, seem to be smaller than liposomes. A large number of studies have shown that there are various proteins, lipids and RNAs on the surfaces of or inside exosomes (Figure 2). Proteins on the surfaces of exosomes include membrane fusion proteins (Rab5,7, Rap1B and annexins I, II, IV, V and VI), antigen presenting proteins (MHC-I and MHC-II), costimulatory molecules (CD86), MVB formation proteins (Gag, Tsg10 and Alix), integrins (α4β1), immunoglobulin-family members (ICAM1) and transmembrane molecules (CD13). Tetraspanins (CD9, CD63, CD81 and CD82) are also abundant on the surfaces of exosomes, and are regarded as special markers of exosomes. Heat shock proteins (Hsc70 and Hsp90), cytoskeleton proteins (Cap1, radixin and advillian) and enzymes (ATPase, glucose 6 and pyruvate kinase) are abundant inside exosomes [13,14]. The lipids include ceramide (sometimes used as a marker to distinguish exosomes from lysosomes), cholesterol, sphingolipids, glycerophosphate, etc., which are generally on the surfaces of exosomes [6]. In addition, the RNAs carried in the exosomes include messenger RNAs (mRNAs), microRNAs (miRNAs) and other non-coding RNAs [15]. Exosomes play an important role in cell–cell communication by transferring proteins, RNAs and lipids on their surfaces or inside themselves to target cells.

### 2.3. The Isolation of Exosomes

In order to be brought into clinical settings, exosomes need to be effectively isolated from various cell fragments and interfering components. In the past, exosomes were often isolated by a method based on ultracentrifugation, which is the most common. However, exosomes produced by ultracentrifugation usually contain microbubbles or other impurities of cell debris. In addition, a high centrifugal force and centrifugal duration may cause damage to exosomes [16]. Therefore, alternative isolation methods based on size, immunoaffinity capture and exosome precipitation have been developed [16,17,18] (Table 1). Microfluidic acoustic, electrophoretic and electromagnetic techniques have been used in recent years to isolate exosomes. The advantages of these methods include great reductions in the required sample capacity, reagent consumption and isolation time [17,18]. However, obtaining large amounts of highly purified exosomes is still a challenge. This may be due to the complexity of the biological fluids from which exosomes are isolated and the similarities in physicochemical properties among exosomes and other EVs [19,20]. Therefore, it is crucial to continuously explore new isolation methods that can obtain more purified exosomes for clinical use.

## 3. The Roles of Exosomes in the Pathogenesis and Progression of Multiple Myeloma

### 3.1. The Promotion of Angiogenesis

Angiogenesis is a hallmark in the progression of MM. It has been shown that the degree of angiogenesis is related to the disease progression and prognosis [30]. Wang [31] et al. demonstrated that MM-derived exosomes directly stimulate the growth, proliferation and invasion of endothelial cells by modulating multiple pathways, including signal transducer and activator of transcription 3 (STAT3), c-Jun N-terminal kinase (JNK), protein kinase B (Akt), p38 and p53-mediated pathways. In addition, they found that many angiogenesis-related proteins are increased in MM-derived exosomes, such as vascular endothelial growth factor (VEGF), basic fibroblast growth factor (BFGF), serpin E1 and angiogenin, which can promote angiogenesis [31,32]. The BM, a highly vascularized tissue, is the main site for exosomes at which to promote angiogenesis. BM is naturally hypoxic compared to other tissues [33]. However, the excessive proliferation of malignant plasma cells makes the BM even more hypoxic, resulting in a situation where MM cells can produce more exosomes. It has been shown that miR-135b is significantly upregulated in the exosomes secreted by hypoxic MM cells. Exosomal miR-135b can promote the formation of endothelial tubes and angiogenesis via the factor-inhibiting hypoxia inducible factor 1 (FIH-1) signaling pathway [34,35].

### 3.2. Immunosuppressive Effects

On the one hand, it is well known that activated myeloid-derived suppressor cells (MDSCs) can exert immunosuppressive effects and promote the growth of cancer cells by suppressing the activation of T lymphocytes [36]. Xiang [37] et al. demonstrated that tumor-derived exosomes can promote the activation of MDSCs, thereby promoting the growth of a tumor. In one study, Wang [31] et al. showed that MM-derived exosomes were able to enhance the expression of the immunosuppressive phenotype of MDSCs by activating the STAT3 pathway, finally inducing suppressive effects on T cells by upregulating nitric oxide synthase (NOS) in the activated MDSCs. On the other hand, it has been shown that natural killer (NK) cells play an important role in the progression of MM. NK cells can be activated at the initial stage of MM and exert a cytotoxic effect, killing MM cells [38]. However, NK cells exposed to the myeloma-derived exosomes show a dose-dependent decrease in their capacity for the specific lysis of MM cells [39].

### 3.3. Promotion of Osteolysis

One of the clinical symptoms of MM is osteolysis, which is induced by an imbalance of bone resorption and formation; specifically, osteoclastic bone resorption is increased, while osteoblastic bone formation is decreased [40,41]. It has been shown that most MM patients manifest osteolytic lesions during the progression of the disease. Osteolytic lesions can induce severe pain and decrease the quality of life for MM patients. On the one hand, there is much evidence showing that exosomes impair bone formation through small non-coding RNAs. Li [42] et al. demonstrated that MM-derived exosomes are enriched in the bioactive lncRNA RUNX2-AS1, which is a product of the RUNX2 antisense strand. The lncRNA RUNX2-AS1 can be transferred to mesenchymal stem cells (MSCs), where it can inhibit the osteogenic activity of MSCs via the exosomal lncRUNX2-AS1/RUNX2 pathway. Moreover, after the exosome secretion inhibitor GW4869 was given to MM model mice, bone loss was reduced and bone formation was maintained. On the other hand, it has also been shown that MM-derived exosomes can induce osteolytic lesions by promoting the differentiation and survival of osteoclasts (OCs). For example, Raimondo [43] et al. demonstrated that MM-derived exosomes are enriched in amphiregulin (AREG), which can increase OC differentiation by activating the epidermal growth factor receptor (EGFR) pathway. Another study demonstrated that MM-derived exosomes are able to promote the migration and growth of pre-osteoclasts by increasing the expression of cysteine X cysteine receptor 4 (CXCR4) and osteoclast markers such as matrix metalloproteinases 9 (MMP9) on pre-osteoclasts [44]. According to research by Faict et al. [45], 5TGM1 cell-derived exosomes can not only promote the differentiation of RAW 264.7 into OCs but also increase the capacity for OC absorption. Meanwhile, they can not only induce the apoptosis of pre-osteoblasts but also inhibit the differentiation capacity of osteoblasts (OBs) in vitro. This is partly due to the transfer of Dickkopf-1 (DKK-1) from exosomes to OBs. After the exosome secretion inhibitor GW4869 was given to MM model mice, Faict et al. found that the inhibition of the secretion of exosomes could reduce osteolytic lesions and furthermore exert an obvious anti-tumor effect when combined with bortezomib [45].

### 3.4. The Roles of Exosomes in Drug Resistance and Survival of Multiple Myeloma Cells

Besides the traditional chemotherapeutic regimen, the novel agents developed for the therapy of MM have significantly improved the therapeutic efficacy in the past few decades, which include IMiDs such as thalidomide and lenalidomide, proteasome inhibitors such as bortezomib and carfilzomib, and monoclonal antibodies such as elotuzumab and daratumumab. IMiDs exert their therapeutic efficacy by inhibiting angiogenesis in MM patients [46]. Bortezomib and carfilzomib target the β5 subunit of the 26S proteasome to treat MM. Elotuzumab and daratumumab exert their therapy efficacy by targeting the signaling lymphocyte activation molecules F7 (SLAMF7) and CD38 on MM cells, respectively [47]. In the past fifteen years, although the survival rate of MM patients has significantly increased due to the application of these effective agents [48], MM is still incurable owing to the rapid development of drug resistance. A widely accepted view is that the drug resistance of MM is induced by complex interactions between the bone marrow microenvironment (BMM) and MM cells [32]. One study has shown that MM-derived exosomes are enriched in miRNA-146a, which can be transferred into MSCs, causing the increased secretion of cytokines and chemokines, including chemokine (C-X-C motif) ligand 1 protein (CXCL1), IL-6, IL-8, MCP-1 and CC chemokine ligand 5 (CCL-5). These cytokines and chemokines enhance the viability of MM cells and promote the migration of MM cells [49]. In a study by Wang et al., exosomes released by bone marrow stromal cells (BMSCs) were demonstrated to significantly improve the viability of MM cells in both the 5T33 murine MM model and human MM cells. BMSC-derived exosomes can also inhibit the effect of bortezomib on repressing the expression of apoptosis-related protein Bcl-2. These both demonstrate that BMSC-derived exosomes play an important role in the drug resistance of MM [50]. Furthermore, Faict et al. [45] also demonstrated that 5TGM1 cell-derived exosomes can increase the viability and survival of MM cells by activating several pathways, including JNK, AKT, p53 and p38. After GW4869 was used to inhibit the secretion of exosomes, they found that the sensitivity of murine MM cells to bortezomib was increased. Another study found that anti-myeloma drugs such as melphalan and bortezomib can induce MM cells to produce large numbers of exosomes, which are called chemoexosomes. The surface of a chemoexosome is enriched with heparanase. When these chemoexosomes are exposed to other MM cells, heparanase can be transferred to other MM cells. This causes the activation of the ERK pathway, the production of TNF-α and the degradation of the extracellular matrix, which may, together, contribute to the development of chemoresistance in MM patients [51,52] (Figure 3).

### 3.5. The Roles of Exosomes in the Proliferation, Homing and Dissemination of Multiple Myeloma Cells

It is well established that BMSCs can promote the growth of MM cells, resulting in MM progression [53,54]. Now, much evidence has shown that MM BMSC-derived exosomes may promote the growth of MM cells and homing to the BM. Roccaro et al. assessed the effect of MM BMSC-derived exosomes on the growth of MM cells both in vitro and in vivo, finding that MM BMSC-derived exosomes can promote the growth of MM cells both in vivo and in vitro. They further investigated whether MM BMSC-derived exosomes could induce cell dissemination and homing to distant BM. Tissue-engineered bones (TEBs) were loaded with MM cells in the presence of MM BMSC-derived exosomes. They were able to visualize the BM vasculature and the presence of MM cells by using in vivo confocal imaging. A higher tumor burden was shown in the mice with TEB implants loaded with MM cells exposed to MM BMSC-derived exosomes than in the mice with implants only loaded with MM cells [55]. In another study, Wang et al. [50] demonstrated that BMSC-derived exosomes of MM patients can promote the proliferation of MM cells, as described in Section 3.4.

## 4. Exosomes as A Diagnostic and Prognostic Tool for Multiple Myeloma

It is known that the disease stage and the probable prognosis of MM patients are very important for choosing the proper therapeutic strategy. Therefore, it is urgent to develop a new non-invasive method that can efficiently monitor the progression of MM [56]. Much evidence shows that exosomes isolated from the peripheral blood of MM patients have the potential to be used as biomarkers for predicting MM progression, as exosomal proteins or miRNAs vary significantly in patients with different prognostic outcomes [57,58,59].

A large amount of evidence has shown that some MM patients have innate resistance or acquire resistance to bortezomib over the course of treatment; drug resistance is why MM remains an incurable malignancy. Therefore, researchers are interested in the early prediction of drug resistance in MM patients. In a study by Zhang et al. [57], 10 miRNAs with the largest changes were overlapped with miRNAs based on literature. These 10 miRNAs were deemed to have the potential to be used as predictive panels for the drug resistance of MM patients. In detail, the upregulation of miR-513a-5p, miR-20b-3p and let-7d-3p, and the downregulation of miR-16-5p, miR-15a-5p, miR-20a-5p, miR-17-5p, miR-125b-5p, miR-19a-3p and miR-21-5p, are involved in the bortezomib resistance of MM patients. Such predictive panels are helpful and important for choosing the most suitable therapeutic regimens for patients.

It has also been demonstrated that the level of circulating exosomal miRNAs is relevant to the prognostic outcomes of MM. In one study, exosomal miRNAs were isolated from the serum samples of 156 MM patients. The qRT-PCR analysis of 22 biologically relevant miRNAs in the samples showed let-7b and miR-18a to be closely correlated with poor outcomes regarding progression-free survival (PFS) and overall survival (OS). Thus, circulating exosomal miRNAs have the potential to be used for predicting the PFS and OS of newly diagnosed MM patients [58].

Furthermore, exosomes may become potential biomarkers for predicting the risk of graft-versus-host disease (GVHD) after ASCT for MM patients. Lia G et al. [59] used flow cytometry to characterize the surface antigens of exosomes in 41 MM patients, thereby finding that the expression of three exosomal surface antigens is related to the onset of acute GVHD. The reduced expression of CD140-α (platelet-derived growth factor receptor) and CD31 (platelet endothelial cell adhesion molecule) is related to a decreased risk of acute GVHD, while the enhanced expression of CD146 (melanoma cell adhesion molecule-1) is correlated with an increased risk of acute GVHD. All the above findings show that exosomes may be used as a prognostic tool for MM in the clinic (Figure 4).

Although, currently, exosomes are mainly promising for MM prognosis, they also show certain potential for the diagnosis of MM (Figure 4). In a study by Sedlarikova et al. [60], 84 types of lncRNAs in exosomes were extracted from the peripheral blood of newly diagnosed MM patients (*n* = 56) and healthy donors (*n* = 36). Firstly, they found that the expression of exosomal lncRNA PRINS was statistically significantly different (*p* = 0.042) between MM patients (*n* = 6) and healthy donors (*n* = 6). Furthermore, they validated this lncRNA PRINS by qPCR in 50 MM patients and 30 healthy donors. The results show that exosomal lncRNA PRINS is able to distinguish MM patients from healthy donors with high sensitivity (80.77%) and specificity (76.92%).

## 5. Therapies Targeting the Secretion of Exosomes

As mentioned before, exosomes play an important role in the BMM that supports the growth and survival of MM cells. Therefore, targeting the secretion of exosomes may become an effective therapeutic strategy for MM (Table 2). It has been demonstrated that the exosome secretion inhibitor GW4869 can not only reduce the osteolytic lesions in MM, but also enhance the sensitivity of murine MM cells to bortezomib, leading to the reversal of drug resistance [42,45]. In addition, it is well established that exosomes are heterogeneous. On the one hand, some exosomes can exert immunosuppressive effects; on the other hand, some can also exert immunomodulatory effects. For example, Vulpis et al. [61] demonstrated that MM-derived exosomes can induce the production of interferon gamma (IFN-γ) in NK cells by activating the NF-κB pathway in a TLR2- and HSP70-dependent manner. Moreover, the genotoxic agent melphalan, at a sublethal dose, can significantly increase the secretion of exosomes from MM cells. Similarly, Borrelli et al. demonstrated that MM-derived exosomes carry the IL-15/IL-15RA complex, which is involved in the proliferation and activation of NK cells. After a sublethal dose of doxorubicin or sublethal dose of melphalan was administered to MM cells, Borrelli et al. found that the drugs could not only cause MM cells to secrete more exosomes but also enhance the expression of the IL-15/IL-15RA complex on the surfaces of MM cells and exosomes via inducing senescence [62]. Therefore, we can conclude that, despite the tumor-supportive and immunosuppressive effects of the exosomes mentioned in the pathogenesis of MM, the immunomodulatory effects of MM-derived exosomes may contribute to improving the NK cell response, which acts at the initial stage of the anti-myeloma response, particularly following intervention with chemotherapy.

## 6. Exosomes as Biological Nanocarriers for Drug Delivery

With the rapid development of drug-delivery technology, nanotechnology is being widely used to deliver drugs. Unfortunately, in spite of the extensive and in-depth investigations on nanocarriers, there has been limited success in clinical applications due to the low safety and low targeting efficiency. Nanomaterials may be cytotoxic. In addition, they can be cleared quickly by the mononuclear phagocyte system (MPS) [63]. Although the PEG modification of nanomaterials can prolong their circulation times in vivo, PEGylation may block the interaction between the nanodelivery system and the target cell, thereby reducing the distribution of the drug in the target cell [64,65]. Inspiringly, exosomes, as trendy biological nanocarriers, have brought new hope to nano drug-delivery systems. As mentioned before, the membranes of exosomes are typical phospholipid bilayers. The membranes of exosomes may directly fuse with the plasma membranes of the target cells, so contained drugs can be directly delivered to the recipient cells. In addition, the membranes of exosomes can well protect the drugs from rapid clearance by the MPS, thereby prolonging the circulation time. Moreover, since exosomes are endogenous biological nanocarriers, they exhibit good safety, biocompatibility and biodistribution compared to liposomes. Owing to these structural characteristics, exosomes may become an ideal biological nanocarrier for drug delivery [66,67,68].

Most chemotherapeutic drugs, such as doxorubicin, have problems with solubility, targeting efficiency and severe adverse effects, which remain their limitations in clinical applications. It has been shown that exosomes modified by targeting ligands can be used therapeutically for the delivery of chemotherapeutic drugs, such as doxorubicin, to tumors, thereby having great potential value for clinical applications. In a study by Tian et al. [69], the tumor targeting ability of exosomes was achieved by engineering immature DCs to express exosomal membrane Lamp2b (lysosome associated membrane glycoprotein 2b) fused with iRGD peptide (CRGDKGPDC). The iRGD peptide specifically combines with αV integrin expressed on breast cancer cells. Then, they obtained purified iRGD exosomes from cell supernatants and loaded doxorubicin into the iRGD exosomes by electroporation. The targeting efficiency and antitumor efficacy of the iRGD-exosomes-doxorubicin were evaluated both in vitro and in vivo. An in vitro experiment showed that iRGD-exosomes-doxorubicin can specifically target the human breast cancer cell line MDA-MB-231, further inhibiting the proliferation of cancer cells. An in vivo experiment showed that iRGD-exosomes-doxorubicin can specifically target the tumor tissue in a nude mouse model of breast cancer, further inhibiting the growth of tumor tissue, without obvious toxicity.

There are still difficulties in the isolation of exosomes and the production yield of exosomes. Therefore, scientists are looking for exosome mimetics (EMs), which are vesicles synthetically isolated from cells. The size and structure of EMs are similar to those of exosomes. In a study by Jiang et al. [70], monocytes (the human U937 monocytic cell line) and macrophages (the mouse Raw264.7 macrophage cell line) were harvested and sequentially extruded through polycarbonate mmbranes (10, 5 and 1 μm) in the presence of doxorubicin, forming doxorubicin-loaded EMs (DOX-EMs). They found that DOX-EMs could target tumor tissue and reduce the growth of tumors in mice bearing subcutaneously transplanted CT26 cells (a mouse colon cancer cell line). In addition, they compared the antitumor activity of DOX-EMs and doxorubicin-loaded exosomes in mice, finding that the DOX-EMs had similar anti-tumor activity to doxorubicin-loaded exosomes. Doxorubicin is a basic drug in the chemotherapy regimen for MM. The main molecules targeting MM cells developed to date are monoclonal antibodies (mAb), including anti-CD38 mAb [71], anti-CD138 mAb [72], anti-BCMA mAb [73] and anti-SLAMF7 mAb [74], which have been used in actual targeted therapy for MM. Based on these developments, the encapsulation of doxorubicin in exosomes or EMs, and the surface modification of exosomes or EMs with targeting molecules (such as anti-myeloma mAbs), may be efficient and promising strategies for the therapy of MM. There are many methods for loading therapeutic agents into exosomes or EMs. These methods are mainly divided into two categories: passive encapsulation and active encapsulation. Passive encapsulations include incubations of exosomes and free drugs [75,76] and incubations of donor cells with free drugs [66]. Active encapsulations include electroporation [77], sonication [78], freeze and thaw cycles [79], click chemistry [80], extrusion [81] and incubation with saponin [82] (Table 3). Notably, these methods result in different encapsulation rates and stability of drugs in exosomes or EMs.

There has been no investigation reporting the application of exosomes in the therapy of MM. However, based on the relevant investigations described above, the encapsulation of anti-myeloma drugs (such as doxorubicin) in exosomes as well as the surface modification of exosomes with targeting molecules (such as anti-CD38 mAb, anti-CD138 mAb and anti-SLAMF7 mAb) are very promising strategies for the therapy of MM. These strategies have the potential to efficiently increase the targeting efficiency of anti-myeloma drugs, reducing their adverse effects and improving their therapeutic efficacy for MM.

## 7. The Roles of Exosomes in Cancer Vaccines

It is well established that immune escape and immunosuppression are essential in cancer pathogenesis. Therefore, immunotherapy has become a potential strategy for cancer therapy, which has shown significant therapeutic efficacy in many clinical trials [83]. The goal of cancer immunotherapy is to stimulate the immune system to recognize and eradicate cancer cells. DCs are important antigen-presenting cells (APCs) that can recognize tumor antigens and present them to T cells, thereby activating specific T cells for an anti-tumor response. Thus, DCs play an important role in cancer immunotherapy. However, DC-based immunotherapy strategies still have some difficulties regarding clinical applications. For example, it is difficult to maintain the efficacy of DCs during long-term storage in vitro. Moreover, when DCs are used in a large number of patients in a clinic, the quality of the DC preparation and the method of DC administration in patients are difficult to standardize [84]. In recent years, DC-derived exosomes (DC-EXs) have been able to overcome these limitations and bring new hope for cancer immunotherapy. It has been observed that DC-EXs can efficiently stimulate an immune response, comparably to the immuno-stimulating ability of the parental DCs [16]. The interactions of exosomes, DCs and other immune cells can regulate the immune responses [85]. Zitvogel et al. [86] showed that DC-EXs can express the MHC-I, MHC-II and T cell costimulatory molecules, which are required for presenting antigens to and activating T cells. In their study, acid-eluted tumor peptide (AEP-P815 or AEP-TS/A)-pulsed DC-EXs were administered to the murine tumor models that were developed by intradermally injecting P815 (an immunogenic but very aggressive mastocytoma) and TS/A (a spontaneous mammary carcinoma) cells into mice. They found that these tumor peptide-pulsed DC-EXs could induce a tumor-specific cytotoxic T lymphocyte (CTL) response in vivo; meanwhile, the growth of the tumor was inhibited in a T cell-dependent manner. Zhen et al. developed ectopic, orthotopic and diethylnitrosamine (DENA)-induced autochthonous hepatocellular carcinoma (HCC) mouse models. They found that α-fetoprotein (AFP)-expressing DC-derived exosomes (DC-EX_AFP_) could induce a specific antitumor response and inhibit the growth of tumors in three HCC mouse models [87]. Escudier et al. loaded MAGE3 antigen into DC-EXs to form a vaccine; then, 15 patients with stage III/IV melanoma who met the inclusion and exclusion criteria were vaccinated four times. No obvious toxicity (>grade II) was observed after vaccination, suggesting that this DC-EX-based vaccine is relatively safe, and it is in a phase I trial for melanoma patients [88]. Similarly, Morse et al. loaded the MAGE tumor antigen into DC-EXs to form a vaccine, which was then administered to 13 patients (nine of them finished this vaccine regimen) with advanced non-small cell lung cancer (NSCLC). The result of a phase I trial showed that this DC-EX vaccine is safe and feasible. Furthermore, the DC-EX vaccine can induce a specific T cell response in some NSCLC patients and prolong the stable period of disease [89]. From the studies with animal models and clinical trials described above, it is apparent that DC-EXs show the potential to be nanocarriers for delivering specific antigens for cancer therapies. Furthermore, DC-EXs themselves can be used as cell-free anti-cancer vaccines to be applied in cancer immunotherapy.

At present, the main specific antigens of MM that show potential in the targeting therapy of MM include BCMA [90], HM1.24 [91], MUC-1 [92], MAGE-C1 [93], B7-H1 [94] and HSP [95]. To date, there has been no investigation reporting the application of DC-EXs in an anti-myeloma vaccine. However, based on the above studies, the encapsulation of MM-specific antigens (such as BCMA, HM1.24 and MUC) in exosomes or DC-EXs with myeloma-specific antigens as cell-free vaccines are promising, and need to be further investigated in the hope of improving the efficacy of therapeutic vaccines for MM. In short, the application of DC-EXs in anti-myeloma vaccines may exert stronger immune responses, accompanied by better safety, biocompatibility [7], stability [96] and biodegradability [97] in vivo (Figure 5).

## 8. Conclusions and Perspective

Although proteasome inhibitors such as bortezomib and carfilzomib, IMIDs such as lenalidomide and pomalidomide, and ASCT have significantly improved the efficacy of MM therapy in recent years, MM is still an incurable hematologic malignancy. It is well established that the tumor-supportive effect of the BMM is the main reason for drug resistance and therapy failure in MM. Exosomes play an important role in the tumor-supportive BMM network; therefore, targeting the secretion of exosomes may become a promising therapeutic strategy for MM. In addition, the abnormal expression of “cargo” (such as miRNAs) in exosomes may be used to help with the diagnosis and prognosis of MM. Therefore, exosomes may become a possible diagnostic and prognostic tool for MM. Additionally, due to the unique nano-level structure of the membranes, along with the safety, biocompatibility, stability and biodegradability of exosomes, the encapsulation of anti-myeloma drugs in exosomes and surface modifications of exosomes with targeting molecules are very promising strategies for the therapy of MM. In addition, MM is characterized by severe immunodeficiency, including a reduction in anti-myeloma immune response, which results in the immune escape and the survival of MM cells. Excitingly, DC-EXs have the potential to be developed into anti-myeloma vaccines to induce stronger immune responses and thus, may eliminate the minimal residual disease (MRD) in MM patients. In fact, there may be a high demand for exosomes in the clinic; however, there is usually only a small number of exosomes separated from cell supernatants. The isolation and purification methods for exosomes are relatively time-consuming and expensive. These factors have brought great challenges to the application of exosomes to cancer therapy. Therefore, the most important problem to solve is developing new methods for producing large amounts of exosomes cheaply. Moreover, exosomes derived from different cells are heterogeneous—some of them can enhance immune responses, while some exert immunosuppressive effects. Therefore, it is necessary to conduct further research on exosomes to ensure their safety and efficacy in clinical applications.

## Figures and Tables

**Figure 1 cancers-13-01635-f001:**
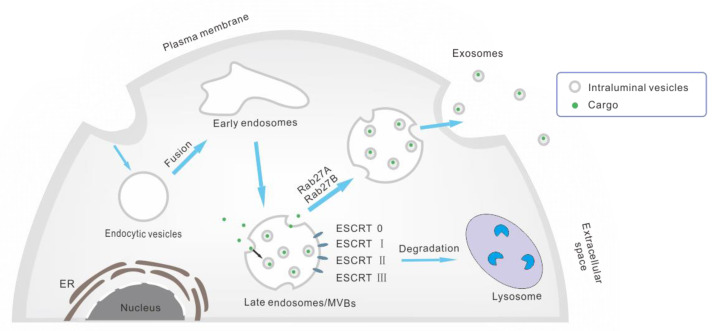
Biogenesis of exosomes. Exosomes are extracellular vesicles of endocytic origin. Early endosomes are formed by the fusion of endocytic vesicles. Then, the inward budding of late endosome membranes leads to the formation of intraluminal vesicles (ILVs). The accumulations of ILVs in the late endosomes are termed multivesicular bodies (MVBs). On the one hand, MVBs may fuse with lysosomes for degradation. On the other hand, with the help of Rab27A and Rab27B, MVBs can transfer to the cellular periphery and fuse with the plasma membrane, releasing ILVs known as exosomes into extracellular space. ER, endoplasmic reticulum; ESCRT, endosomal sorting complexes required for transport; ILVs, intraluminal vesicles; MVBs, multivesicular bodies.

**Figure 2 cancers-13-01635-f002:**
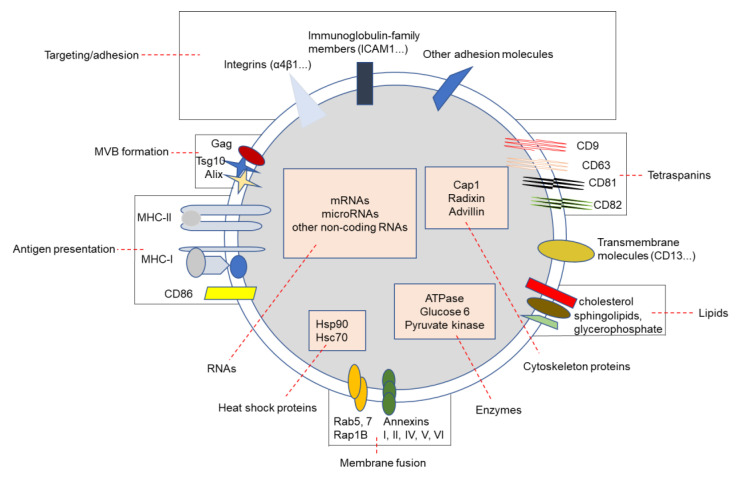
Different components of exosomes. There are various proteins on the surfaces of or inside exosomes, such as membrane fusion proteins, cytoskeleton proteins, tetraspanins, integrins, transmembrane molecules, antigen presenting proteins, costimulatory molecules, MVB formation proteins, enzymes and heat shock proteins. Lipids and RNAs are also enriched on the surfaces of or inside exosomes. These components can mark exosomes or participate in the transport of exosomes for cell–cell communication.

**Figure 3 cancers-13-01635-f003:**
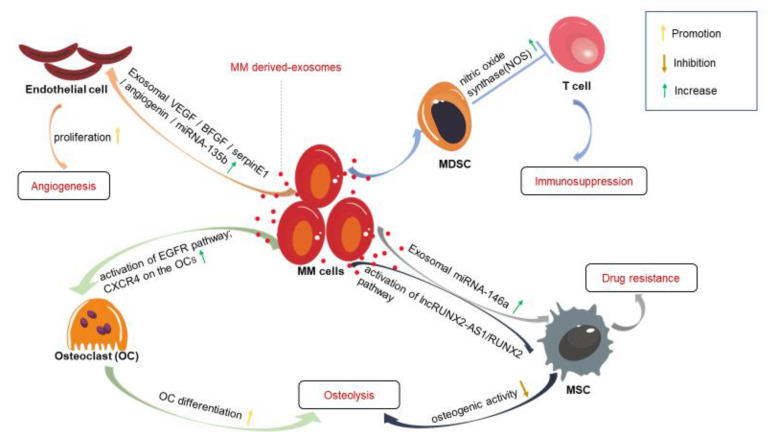
The effects of MM-derived exosomes on different types of cells in the BMM. MM-derived exosomes act on different types of cells in the BMM, thereby creating an environment that benefits the survival and growth of MM cells. MDSC, myeloid-derived suppressor cell; MSC, mesenchymal stem cell; OC, osteoclast; VEGF, vascular endothelial growth factor; CXCR4, cysteine X cysteine receptor 4; BFGF, basic fibroblast growth factor; EGFR, epidermal growth factor receptor.

**Figure 4 cancers-13-01635-f004:**
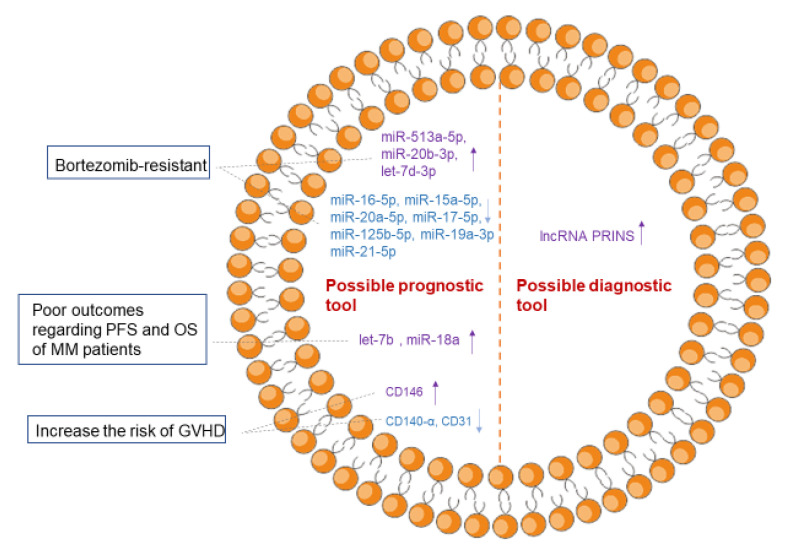
The potential of exosomes in the diagnosis and prognosis of MM. Exosomes may be used as a prognostic tool in MM due to their potential for the early prediction of drug resistance, the outcomes regarding PFS and OS and the risk of GVHD after ASCT. In addition, exosomes may be used as a diagnostic tool for MM owing to the abnormal expression of “cargos” in exosomes. CD146, melanoma cell adhesion molecule 1; CD140-α, platelet-derived growth factor receptor; CD31, platelet endothelial cell adhesion molecule; PFS, progression-free survival; OS, overall survival; GVHD, graft-versus-host disease; ASCT, allogeneic stem cell transplantation.

**Figure 5 cancers-13-01635-f005:**
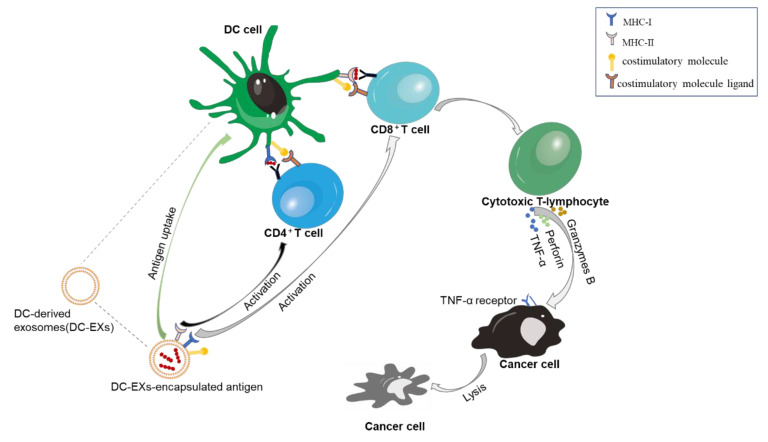
DC-EX-based vaccines induce an anti-cancer immune response in vivo. Cancer-specific antigens are loaded into DC-EXs to form an anti-cancer vaccine. DC-EXs can express MHC-I, MHC-II and T cell costimulatory molecules. Therefore, they can present antigens to CD8^+^ and CD4^+^ T cells, and induce the activation of specific T cells and a strong CTL response. Then, CTLs release granzyme B, perforin and TNF-α to efficiently kill cancer cells. DC-EXs, DC-derived exosomes; MHC-I, major histocompatibility complex class I; MHC-II, major histocompatibility complex class II.

**Table 1 cancers-13-01635-t001:** Comparison of the existing isolation methods for exosomes.

Isolation Techniques	Isolation Methods	Advantages	Disadvantages	Ref
Ultracentrifugation Techniques	Differential Ultracentrifugation	No sample pretreatment required; Suitable for large sample capacity; Wide application.	Time-consuming; Low purity; High-speed centrifugation may damage exosomes.	[21]
	Density Gradient Centrifugation	Effective separation of exosomes from protein aggregates.	Low production; High equipment cost; Cumbersome.	[15]
Size-Based Techniques	Ultrafiltration	Fast; Low cost; No requirement for special equipment; This method may directly extract RNAs from exosomes.	Some of the exosomes may be left on the filter membrane; Exosomes may be lysed due to shear force.	[22,23,24]
	Exosome Isolation Kit	The operation process is simple.	Some of the exosomes may remain on and block the filter membrane.	
	Sequential Filtration	Automatable; The integrity and biological activity of exosomes are not affected.	Some of the exosomes may remain on and block the filter membrane.	
	Size-Exclusion Chromatography	Highly purified exosomes; The integrity and biological activity of exosomes are not affected.	Time-consuming.	
	Flow Field-Flow Fractionation	Novel technique developed recently; Superior reproducibility; Gentle and fast.	Not mentioned.	
Immunoaffinity Capture-Based Techniques	ELISA	Highly purified exosomes; This method is excellent for the isolation of specific exosomes.	Sample pretreatment is required Low production.	[25]
	Magneto-Immunoprecipitation	Highly purified exosomes; Low cost on equipment; Suitable for large sample capacity; High efficiency.	High reagent cost, The antigenic epitope may be blocked.	[15] [26]
Exosome Precipitation	Polyethylene Glycol (PEG) Precipitation	Fast; The operation technique is simple.	Exosomes, extracellular proteins and other EVs may be precipitated together.	[27]
	Lectin-Induced Agglutination	Simple; Highly purified exosomes.	Sample pretreatment is required.	[28]
Microfluidic-Based Isolation Techniques	Acoustic Nanofilter	Less starting volume of sample; Fast and simple.	Lack of method validation.	[18]
	Immuno-Based Microfluidic Isolation	Less starting volume of sample; The highest cost efficiency while the smallest amount of time.	Lack of method validation.	[29]

**Table 2 cancers-13-01635-t002:** Therapies targeting the secretion of exosomes.

Intervention Targeting Exosome Secretion	Exosome Secretion	Outcomes after Targeting the Secretion of Exosomes	Ref
Treatment with exosome secretion inhibitor GW4869	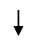	Reduction in osteolytic lesions;Improvement of the sensitivity of murine MM cells to bortezomib;Further reversal of drug resistance of MM.	[42][45]
Treatment with sublethal dose of melphalan	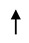	Increased secretion of MM-derived exosomes, which can induce the production of IFN-γ in NK cells.	[61]
Treatment with sublethal dose of melphalan or doxorubicin	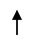	Promotion of the secretion of exosomes by MM cells;Enhancement of the expression of IL-15/IL-15RA complex via inducing senescence.	[62]

**Table 3 cancers-13-01635-t003:** Comparison of different methods for loading therapeutic agents into exosomes or exosome mimetics.

Loading Categories	Method Classification	Loading Process	Advantages	Disadvantages	Ref.
Passive loading	Incubation of exosomes and free drugs	Exosomes are incubated with drugs, and then, the drugs can interact with the lipid layer of the vesicle membrane and diffuse into the exosomes.	Simple for preparation; The structural integrity of exosomes is not affected.	Low drug-loading efficiency.	[75,76]
	Incubation of donor cells with free drugs	Donor cells are incubated with drugs, and then, donor cells secrete exosomes that encapsulate the drugs.	Simple for preparation; The structural integrity of exosomes is not affected.	Low drug-loading efficiency; The proliferation capacity of donor cells may be affected.	[66]
Active loading	Electroporation	Electrical field creates small pores in the exosome membrane, and then, drugs or nucleotides can subsequently diffuse into the interior of the exosomes via the pores.	This method can load large molecules.	This method may cause the aggregation of RNA; Low drug-loading efficiency.	[77]
	Sonication	The mechanical shear force from the ultrasonic probe compromises the membrane integrity of the exosomes, and then, drugs diffuse into the exosomes.	The content of membrane-bound proteins or lipids in exosomes is not changed; High drug-loading efficiency.	Some drugs may attach to the outer layer of the membrane.	[78]
	Freeze and thaw cycles	Drugs are incubated with exosomes at room temperature for some time. Then, they are quickly frozen at −80 °C or in liquid nitrogen and thawed at room temperature. This process is repeated at least 3 times.	The drug-loading efficiency is higher than that in passive incubation.	The dilution ratio of lipid may be affected.	[79]
	Click chemistry	Drugs can be directly attached to the surface of the exosome through covalent bonds.	Fast and efficient; This is an ideal method for the attachment of small molecules and macromolecules to the surface of exosomes.	Not mentioned.	[80]
	Extrusion	The membrane of exosomes is extruded and compromised by a syringe-based lipid extruder. Then, drugs diffuse into exosomes.	High drug-loading efficiency.	The membrane integrity of exosomes may be damaged.	[81]
	Incubation with saponin	Saponin can form complexes with cholesterol in cell membranes, thus generating the pores. Then, drugs diffuse into exosomes.	Saponin does not degrade the catalase in exosomes.	Saponin may cause hemolysis in vivo.	[82]

## Data Availability

No new data were created or analyzed in this study. Data sharing is not applicable to this article.

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
