# Peer review of "Recent Progress of Exosomes in Multiple Myeloma: Pathogenesis, Diagnosis, Prognosis and Therapeutic Strategies"

_cancers, 2021, doi:10.3390/cancers13071635_

Round 1

Reviewer 1 Report

In their review entitled “Recent progress of exosomes in multiple myeloma:pathogenesis, diagnosis, prognosis and therapeutic strategies” Xi Wang et al discuss exosomes, their isolation, their role in MM pathogenesis, their potential as a diagnostic tool or for targeting in MM generally and more specifically in cancer vaccines.

The topics covered by the authors are timely and of interest to a wide audience.  The authors have covered a significant amount of ground in this manuscript and the figures and tables re-enforce the key messages.

Although the narrative and content that the authors wish to convey is evident, and a substantial amount of work has been undertaken to draft the manuscript, there is a significant issue with the written English expression throughout the manuscript.  The underlying message is good, but the grammar/syntax is not of scholarly quality and will require improvement.

Here are just half a dozen examples of errors, which are constant throughout the text.  They undermine the effort undertaken in the assembly of this paper.

1

Ln 25 (first sentence of abstract)

Multiple myeloma (MM) is one of the hematological malignancies, it is still incurable

Should be

Multiple myeloma (MM) is one of the hematological malignancies that is still incurable

2

Ln 30

released by a variety of cells. And some exosomes large

Avoid starting a sentence with the word ‘And’.  There are several instances of this throughout the text.

3

Ln 33

important roles of exosomes in BMM, targeting

An article (a/the) is needed before BMM used in this way.

4

Ln 35

diagnose MM or early predict the prognosis

‘Early predict’ is very awkward phrasing.  Suggest ‘diagnose MM or used as part of a screen for the early prediction…’

5

Ln 38

surface modification of exosomes with the targeting molecule are very promising strategies

Should be

surface modification of exosomes with targeting molecules are very promising strategies

6

Ln 69

‘size, biocompatibility and safety, exosomes in some of the current researches perform…’

research performs

Table 1

It is suggested to rotate Table 1 by 90 degrees so that the text is less cramped.

Table 3 would also benefit from rotation.

‘Proved’

In a few places in the manuscript the authors use the phrase ‘proved’.  It might be better to use the phrase ‘demonstrated’ or ‘observed’.

Ln 316

Interferon gamma is usually abbreviated IFN-γ

MPS

The abbreviation MPS was used twice but I was not able to find it defined.

Mononuclear phagocyte system?

Avoid where possible using abbreviations in headings.

Reviewer 2 Report

This review by Wang X. et al focuses on the relevat issue of the role of exosomes in multiple myeloma, ranging from their  role on pathogenesis,  progression and resistance to therapy  of disease to their potential clinical application. Although very thorough, some sections require substantial revision in order to be less dispersive and more focused on multiple myeloma.  In detail

Section 4: I do not think that we need exosomes for diagnosis of myeloma and their impact on this issue should be smooth

Section 6: It should be considerably reduced focusing  on MM drug delivery.

Section 7: Need consistent revision, by shortening the general considerations and being more detailed on available experimental models.

Minor

Never start a sentencece with “And”

Reviewer 3 Report

I reviewed the manuscript of entitled "Recent progress of exosomes in multiple myeloma: pathogenesis, diagnosis, prognosis and therapeutic strategies". My comments are below.

They describe the review article based on a lot of evidence in the citation. It will be interesting for readers.

Author Response

Reviewer 3 didn't show detailed comments on our manuscript.

Reviewer 4 Report

In this manuscript, Wang et al. first review the main characteristics of exosomes and their biogenesis as well as a comparison of the existing methods for their isolation. Next, they go through the main roles reported for exosomes in the pathogenesis of multiple myeloma (MM): specifically, angiogenesis, immunosuppression, osteolytic lesions, drug resistance, and survival. Finally, they go over the potential use of exosome components for myeloma diagnosis and prognosis, the possibility of targeting exosome secretion as a therapeutic approach in MM, and the potential use of exosomes as nanocarriers for drug delivery or the use of dendritic cell-derived as vaccines for MM. 

Although the review is well organized and keeps a good structure, major concerns have been found that preclude its publication at its present form and should be carefully addressed and corrected by the authors:

  1. The review needs an extensive review of the English language and style. Specifically, many grammar errors together with essay writing along the whole text require editing. Some words also need checking for correct meaning.
  2. Although the role of exosomes is reviewed in several aspects of MM pathogenesis, their contribution to myeloma cell growth/proliferation, and in homing and dissemination in the bone marrow is missed.
  3. In sections such as exosomes to be used as nanocarriers for anti-myeloma drugs or their use as vaccines, authors are encouraged to clearly define which reports refer to approaches in other tumors and which refer to MM. Besides, authors are suggested to give some ideas for the implementation of these approaches to the treatment of MM.  
  4. The manuscript also needs a thorough review for accuracy. Some items are listed below, but the whole manuscript should be revised:

- ln 79: check for “intracellular substances”

- ln 85: check for definition of ESCRT complexes

- Figure 1 needs correction. Specifically, check on plasma membrane/vesicle membrane depiction. The formation of intraluminal vesicles is not correctly represented.

- Please improve the description of the molecular components of exosomes. Maybe relating them to whether they are components of the exosome membrane or carried inside the exosome. Also “sugar chains” may be reconsidered to be part of “proteoglycans”. Also check Figure 2 for the appropriate location of some represented components: Annexins, lipids, MCB formation proteins. Also, check for overlayered letters in tags.

-  ln 121: “label”, “transporting”

- Table 1 needs review: “Ultrafiltration” and “Exosome isolation kit” within ultracentrifugation techniques; check whether “special equipment is required” for size exclusion chromatography, or “high cost on equipment” for PEG precipitation. Correct “Widely”, “Effectively”. Please also use lines or different shadings to separate rows, since these are difficult to follow.  

- ln 161: “bone”

- ln 174: please cite the original reference, since ref 39 is a review

- ln 189 and 195:  authors and references 43 and 44 are interchanged

- Figure 3: authors may want to differentiate between exosome content and the actual effect that exosomes cause on target cells (e.g.NOS ↑, CXCR4 ↑, miR-146a↑)

- ln 249: references 54-56 are general references of exosomes as biomarkers for tumor progression or prognosis or response, but not in MM

- ln 254: it is my understanding that changes in exosomal miRNAs in different stages of MM has not been shown in refs 57 and 58

- pages 14, 15 and 16: in my opinion, it is a wrong assumption that because exosomes from myeloma cells or mesenchymal stromal cells (MSCs) may have been found to have special traits (e.g. high IL6, CCL2, fibronectin) in exosomes from MSCs, these may become biomarkers for diagnosis or prognosis. Please modify Fig 4 accordingly

- page 17: it is assumed by authors (ln 325) that treatment with melphalan or doxorubicin (refs 63 and 64) may be used to increase NK immunoreactivity against myeloma cells. However, it is of note that: i) in refs 63 and 64 cells are treated only with sublethal concentrations of melphalan or doxorubicin;  ii) NK cells (except for initial stages) in MM are not immune responsive;  iii) myeloma cell-derived exosomes despite activating NK cells may also contain many factors which may promote tumor growth

- ln 350: explain the type of cell (EL-4)

- ln 363: please explain first the purpose of the whole description of modified exosomes in breast cancer: “exosomes modified by targeting ligands can be used therapeutically for the delivery of drugs such as doxorubicin to tumors, thus having great potential value for clinical application”. Also, explain iRGD peptide

- ln 377: explain bEND3

- page 20: earlier examples of the use of exosomes to cross the blood brain barrier were reported (e.g. Alvarez-Erviti L, Seow Y, Yin H, Betts C, Lakhal S, Wood MJ. Delivery of siRNA to the mouse brain by systemic injection of targeted exosomes. Nat Biotechnol. 2011 Apr;29(4):341-5. doi: 10.1038/nbt.1807. PMID: 21423189). Nevertheless, does this issue have any relevance for the treatment of MM?

- ln 386: explain what an “exosome mimetic” is and how it is made

-Table 2: please use lines or shading to delimit rows in the table. “Antibody binding” (ref 81) is not a drug delivery method, but a method for monitoring exosomes

- ln 428:   which antigen was loaded?

- ln 433: which were the results from the trial in ref 89 (2005)

- ln 441: which vaccine are the authors referring to?

- ln 465: which MM specific antigen?

  1. Please explain acronyms the first time they appear in the text (e.g. MPS, RES...)
  2. Authors are encouraged not to use the “Dex” acronym for dendritic cell-derived exosomes since it is coincident with the abbreviation for “Dexamethasone”, one of the currently used drugs in MM treatment.

7. Authors´ contributions may be summarized. 

Round 2

Reviewer 1 Report

The authors have extensively modified the manuscript.  The written expression requires substantial further improvement.  The underlying message is sound and the manuscript will be helpful for scholars with an interest in this field but it requires editing.

A minor point (among many) is the use of the word 'researches'.  Research is not a countable noun and so the use as a plural (eg 'We did 3 researches') is not correct ('we did 3 research projects').  Some dictionaries may list 'researches' as a noun but it is at best archaic.

Reviewer 2 Report

-

Author Response

The reviewer didn't give the detailed comments on our manuscript.

Reviewer 4 Report

Authors have addressed most of the comments and suggestions from the first review, as well as corrections. Nevertheless, modifications and checkings are still needed and only some of them are highlighted bellow. Careful reading for grammatical, essay and use of the English language is encouraged.

-  ln 35 “this review worth the concern”: check for grammar and meaning

-  ln 47 “therapy strategy”: therapeutic strategy?

- ln 49 “prediction of the prognostic outcomes”

- ln 54 “Therefore, DC-EXs may be used as the nanocarrier to deliver cancer vaccines”: check as Therefore, DC-EXs may be used as a nanocarrier to deliver cancer vaccines in MM?

- ln 66 “Medulla-supressed anemia”: please check

- ln 75: “As is known, that cells will secrete various types of extracellular vesicles (EVs),.” Please check

- ln 77-78: “The diameter of apoptotic bodies and microvesicles are relatively large”

- ln 107: Authors have made some corrections in the text. However, although changes have been made in Fig 1, still it is my opinion it does not represent the view for the formation of intraluminal vesicles (ILVs) by inward budding of the membrane of the multivesicular body (MVB). I would not depict any vesicles inside the early endosome and then ILV forming at late endosome/MVB should have the same membrane as that of the MBV (not orange but black) and the inside would be white/light gray, the same as the cytoplasm. If cargo wants to be represented inside the ILVs, then some cargo may also be portrayed in the cytoplasm at the indentations of late endosome/MVB.

- Table 1: Please check whether “Sample pretreatment is required” is correct for PEG precipitation

Also check for “More geniality to avoid the structure damage of the exosomes”in flow field-flow fractionation.

- ln 215-216 “Whereas, they can not only induce the apoptosis of osteoblast (OB) but also inhibit the differentiation of OB” check your essay

- ln 251-252 “When these chemoexosomes are exposed to other MM cells, heparinase will transfer to other MM cells”. Please check “heparinase”

- ln 347-348 “Moreover, the genotoxic agent melphalan with the sublethal dose can significantly increase the secretion”. Please check for use of English language.

- ln 351: “After sub-lethal dose of doxorubicin or melphalan are given”. Please check for use of English language.

- ln 389 “with av integrin” should be “with α5 integrin”

- ln 400 “Raw264.7 macrophages cells” should be “Raw264.7 macrophage cell line”
